

# Innovative utilization of argon plasma coagulation combined with endoclips for managing gastrointestinal bleeding attributed to colonic diverticular bleeding: a retrospective study

Zihan Huang[*], Xiaomeng Feng[*], Xin Yin, Tao Sun, Chongxi Fan, Hongyu Chen, Bairong Li and Shoubin Ning

Department of Gastroenterology, Air Force Medical Center, Air Force Medical University, Beijing, China
[*] These authors contributed equally to this work.

## ABSTRACT

**Background**. Colonic diverticular bleeding is one of the primary causes of lower gastrointestinal bleeding, with endoscopic hemostasis as the first-line treatment. However, the outcomes of endoscopic treatments remain suboptimal. This study utilized an innovative therapeutic method to manage colonic diverticular bleeding and evaluated its feasibility and safety in clinical settings.

**Methods**. Between July 2018 and July 2023, 35 patients with colonic divercular bleeding were treated through argon plasma coagulation combined with endoclips (APC-EC) at the Air Force Medical Center. The technical success rate, rebleeding rate, and complications associated with the therapeutic method over a 5-year period were retrospectively analyzed.

**Results**. The technical success rate of the method was 100%, the early rebleeding rate of APC-EC was 8.57%. The incidence of late rebleeding at 1-year follow-up was 5.71%, it was 0% at the 2- and 3-year follow-up periods. Intraoperative perforation was observed only in two patients treated with APC-EC; however, delayed perforation was not observed in any patient, and none of them required surgical treatment.

**Conclusions**. APC-EC might be a viable, safe, and effective method for treating colonic divercular bleeding.

Corresponding authors
Bairong Li, airbai8040@163.com
Shoubin Ning, ningshoubin@126.com

## INTRODUCTION

Colonic diverticular bleeding (CDB) accounts for approximately 40% of cases involving serious hematochezia and is among the main causes of lower gastrointestinal bleeding. Colonoscopy serves as the primary diagnostic tool in presumptive cases of acute lower gastrointestinal bleeding or instances related to CDB (*Longstreth, 1997*; *Strate & Gralnek, 2016*; *Nagata et al., 2019*). Endoscopic hemostasis has emerged as the first-line treatment for managing CDB (*Tsuruoka et al., 2020*). Definitive diagnosis of CDB based on signs of

recent hemorrhage (SRH) through colonoscopy poses a significant challenge because of a low detection rate, ranging from 19% to 36% (*Sugiyama et al., 2015*; *Sugihara et al., 2016*). Moreover, endoscopic hemostasis cannot be performed until the source of bleeding is unknown. Approximately 70%–80% cases of CDB can be managed through conservative treatment; however, rebleeding is observed in nearly 25% of these cases (*Longstreth, 1997*; *McGuire Jr, 1994*). Patients experiencing rebleeding may require repeated hospitalizations, blood transfusions, and examinations, which adversely affect their overall quality of life. In addition, colectomy is required for managing patients with persistent, massive, and acute bleeding accompanied by hemodynamic instability (*Sugihara et al., 2016*). Therefore, in clinical practice, CDB treatments are not generally minimally invasive and aim to achieve hemostasis. Traditional endoscopic hemostasis methods have been shown to be effective in CDB management; however, a standard treatment for CDB with negative SRH is lacking, and the rate of rebleeding after endoscopic treatment remains inconsistent (*Tomizawa & Strate, 2022*). Additionally, endoscopic hemostasis methods are ineffective in preventing CDB recurrence in the long term (*Nagata et al., 2019*). Considering these unexplored aspects, in this study, we used a new endoscopic hemostasis method, argon plasma coagulation combined with endoclips (APC-EC), for the treatment of CDB patients with or without SRH. The feasibility, efficacy, and safety of this method were comprehensively assessed to provide a foundation for its subsequent clinical applications.

## PATIENTS AND METHODS

This study was designed as a 5-year retrospective cohort study and conducted in patients with CDB who underwent APC-EC. Patients fulfilling the following criteria were included: (1) CDB presenting with bright or dark red bloody stools; (2) definitive or presumptive diagnosis of diverticular bleeding through an endoscopic examination; (3) having undergone APC-EC; and (4) the follow-up period of at least 1 year. The definitive diagnosis of diverticular bleeding was based on SRH including active bleeding, non-bleeding visible blood vessels, and adherent clots that developed into active or non-bleeding visible blood vessels after clot removal. The presumptive diagnosis of divercular bleeding was based on colonoscopy observations, including the lack of evidence for bleeding from the diverticulum or other major colonic lesions and a negative result in small bowel evaluation (*Jensen et al., 2016*). Diverticula suspected as having a high bleeding risk were characterized by the presence of dilated and tortuous blood vessels at the bottom, neck, or around the opening of the colonic diverticula. Small bowel bleeding was precluded with computed tomography (CT), CT enterography, double-balloon endoscopy (DBE), or capsule endoscopy (CE).

Patients experiencing bleeding from other causes and those with incomplete or missing clinical data were excluded. All patients signed a written informed consent form and were followed-up until July 2024. The patients' clinical symptoms after APC-EC treatment; timing, etiology, and treatment of rebleeding; and changes in hemoglobin were assessed during the follow-up. This study was approved by the Ethics Committee of the Air Force Medical Center (IRB: PLA [New Technology] No. 2022-17-PJ01).

The early rebleeding and late rebleeding rates after APC-EC treatment were considered as the primary outcomes. Early and late rebleeding were defined as rebleeding occurring

within and after 30 days of the initial treatment, respectively (*Kobayashi et al., 2020*). Technical success was defined as successful completion of the 3-step APC-EC process. The patients' characteristics, clinical outcomes, and adverse reactions were considered as the secondary outcomes. SPSS 25.0 software was used for statistical analysis (IBM Corp., Armonk, NY, USA).

## Colonoscopy and equipment

All patients received routine medical treatments, including monitoring of vitals, fluid resuscitation, and blood transfusion, depending on the grade of gastrointestinal bleeding. According to the patients' general condition, the examining physician determined whether polyethylene glycol solution or saline enema should be used for bowel preparation before colonoscopy. Contrast-enhanced CT or CT enterography was performed to identify the bleeding site in each patient, unless the cause of bleeding had been identified through urgency colonoscopy. CE or DBE was performed to exclude small bowel bleeding. The following equipment were used to perform APC-EC: colonoscope, water jet system (CF-H290I or PCF-Q260 AZI; Olympus Optical Co, Ltd, Tokyo, Japan), transparent hood (MAJ663; Olympus), rotatable clips (ROCC-D-26-230; Micro-Tech Co., Ltd, Nanjing, China), high-frequency generator (Erbotorm ICC200; ERBE, Tubingen, Germany), and APC probe.

## APC-EC hemostasis procedure

The process of endoscopic hemostasis involved the following steps: (1) Identification and cleansing of the diverticulum: A colonoscope with a transparent hood and water jet system was inserted to locate the SRH-positive or suspected diverticulum. The transparent hood enabled observation of the diverticular dome by expanding the visual field. Its suction power was utilized to invert the diverticulum or an APC tube was employed to remove the remaining debris, if any. The diverticulum was thoroughly cleansed with flowing water, and fecal matter was flushed out from its cavity to ensure cleansing at its base. In case bleeding had ceased, we determined whether the diverticulum is SRH-positive by using device attachments, such as a transparent hood or water jet, for mucosal suctioning or flushing. (2) APC: An APC catheter was inserted, and the probe tip was placed at 2–6 mm distance from the bleeding lesion. APC was performed in blood vessels of the dome and neck mucosa of the diverticulum, resulting in sufficient coagulation of the tissues surrounding the diverticulum and changes in the color of the APC areas concentrically from black in the center to yellow and white on the outer edge. (3) Endoclip placement: The area of tissue coagulation was firmly closed using endoclips in a zipper manner. Figure 1 provides an overview of the APC-EC procedure.

All endoscopic procedures were performed by endoscopists with expertise in emergency colonoscopy. Specifically, all endoscopists involved in this study had performed more than 3,000 colonoscopies and accumulated extensive experience in endoscopic treatment.

## Further treatment after initial endoscopic treatment

After endoscopic treatment, the patients were asked to fast for at least 48 h. Colonoscopy was performed immediately in patients who experienced rebleeding to determine the

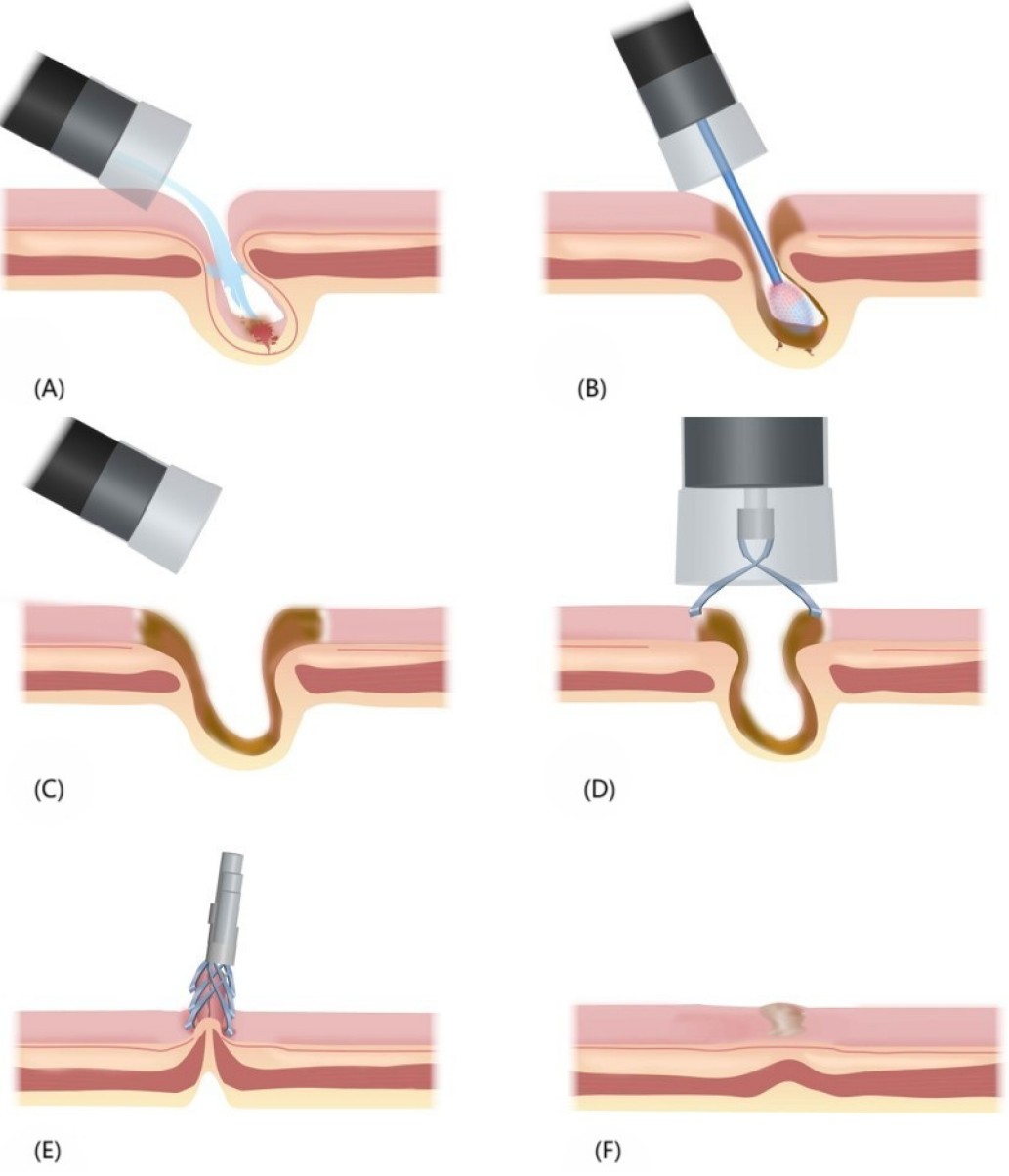

**Figure 1** **Overview of the APC-EC hemostasis process.** (A) Insertion of a colonoscope with a transparent hood and water jet system to locate the SRH-positive or suspected diverticulum. (B) Insertion of an APC catheter, and execution of APC in the dome and neck mucosa of the diverticulum. (C) The fully coagulated diverticulum. (D) Suctioning of intestinal gas to shrink the wound, and endoclip placement for the sealing of wound. (E) Multiple endoclips arranged in the form of zippers, which facilitated wound closure. (F) Healing of the surface and scar formation after endoclip detachment.

rebleeding site. For patients with multiple diverticula and presumptive divercular bleeding, multiple APC-EC treatments were required, with an interval of at least 3 months between consecutive treatments, until all diverticula with a high risk of bleeding are treated.

## RESULTS

A total of 35 patients (average age: 58.3 ± 10.7 years (range, 43–94 years)) with diverticular bleeding were treated using APC-EC. Their characteristics are presented in Table 1. Male patients accounted for 91.4% of the total. Overall, the patients were overweight, with an average body mass index (BMI) of 25.6 ± 2.9 kg/m$^2$. Main comorbidities of the patients were hypertension (77%), diabetes mellitus (31%), and cerebrovascular diseases (20%). In addition, 20% of the patients had been taking oral antiplatelet drugs. All patients with suspected small bowel hemorrhage, except those with definite diverticular bleeding found in initial colonoscopy, had their small bowels examined. Of the total patients, 88.57% (31/35) underwent CT or CT enterography, 31.43% (11/35) underwent digital subtraction angiography, 22.56% (8/35) received CE, and 88.57% (31/35) were evaluated through DBE. DBE was performed *via* the anal route in 29 patients and through the oral route in 20 patients.

The data indicated that the patients exhibited remarkable differences in their disease course. CDB course exceeded three months in 60% of the patients and even over 10 years in others. About two-thirds of the patients had a history of recurrent gastrointestinal bleeding, with an average of five bleeding episodes. Additionally, 71.8% of the patients experienced hemorrhagic shock, and 80% required blood transfusion.

Hemostasis could be achieved in all patients immediately after the initial endoscopic treatment through APC-EC. Among the 35 patients, 12 (34.29%) had definite diverticular bleeding, which included active bleeding ($n = 5$), a non-bleeding visible vessel ($n = 3$), and an adherent clot ($n = 4$). Additionally, 23 (65.71%) patients were diagnosed as having presumptive diverticular bleeding after excluding other causes of gastrointestinal bleeding. Within 24 h of admission, an emergency colonoscopy was performed in 19 cases, of which seven cases were treated without the oral bowel preparation. Among these, 10 cases showed signs of CDB under endoscopy. CDB was diagnosed through emergency colonoscopy in 10 of the 19 patients (52.63%).

A total of 59 diverticulum treatments were performed through colonoscopy on the 35 patients. Specifically, 62.86% (22/35) of the patients underwent 2–3 prophylactic diverticulum ablations, each with over 3-month interval. A total of 425 colonic diverticula were treated, averaging 12.14 ± 10.67 per patient and 7.2 ± 6.0 per colonoscopy session. Figure 2 shows changes in the levels of inflammatory markers. On day 4 of the treatment, the white blood cell (WBC) count returned to its baseline level. C-reactive protein (CRP) levels peaked on days 2 and 3, followed by a gradual decrease on day 4. Follow-up colonoscopies demonstrated the formation of healing scars and granulomas in the treated patients. Six months after the treatment, the patients' average hemoglobin increased significantly by 42.91 ± 29.47 g/L.

Three (8.57%) patients suffered early rebleeding after the initial APC-EC treatment. Two patients with presumptive diverticular bleeding experienced early rebleeding within 9–10 days of APC-EC because of insufficient clamps, which necessitated endoclip detachment and resulted in bleeding from the ulcer. The cases of rebleeding were treated using endoclips. One patient with definite diverticular bleeding experienced early rebleeding within four

**Table 1  Characteristics of 35 patients with colonic diverticular bleeding.**

| Characteristic | Value |
| --- | --- |
| Age, y, mean ± SD | 58.31 ± 10.68 |
| Sex, male | 32 (91.4%) |
| BMI, kg/m$^2$, mean ± SD | 25.59 ± 2.94 |
| Comorbidity | |
|     Hypertension | 27 (77.14%) |
|     Diabetes mellitus | 11 (31.43%) |
|     Cerebrovascular disease | 7 (20%) |
|     Coronary heart disease | 5 (14.59%) |
|     Chronic kidney disease | 2 (5.71%) |
|     Chronic obstructive pulmonary disease | 1 (2.86%) |
|     Prostate cancer | 1 (2.86%) |
| Medication history | |
|     Antiplatelet drugs | 7 (20%) |
|     Anticoagulant drugs | 1 (2.86%) |
|     Steroid | 1 (2.86%) |
| CT/ CT enterography examination | 31 (88.57%) |
| Digital subtraction angiography | 11 (31.43%) |
| Capsule endoscopy | 8 (22.56%) |
| Double-balloon endoscopy | 31 (88.57%) |
|     Oral | 20 |
|     Anal | 29 |
| Course of disease | |
|     >1 year | 16 |
|     >3 months | 5 |
|     <10 days | 14 |
| History of diverticular hemorrhage | 22 (62.86%) |
| History of hemorrhagic shock | 25 (71.83%) |
| History of blood transfusion | 28 (80%) |
| Colonoscopy | |
|     Emergency colonoscopy (<24 h) | 19 |
|     Definite diverticular bleeding | 12 (34.29%) |
|     Presumptive diverticular bleeding | 23 (65.71%) |
|     Stigma of hemorrhage | |
|         Active bleeding/non-bleeding visible vessel/adherent clots/ | 5/3/4 |
|     At least 2 colonoscopies | 22 (62.86%) |
| Total number of colonic diverticulums treated | 425 |
| Hemoglobin growth, g/L, mean ± SD | 42.91 ± 29.47 |

days of the treatment, attributable to insufficient carbonization of the diverticular vessel. Hemostasis was achieved after the re-application of APC-EC.

All patients were followed-up either telephonically or through their visit to the outpatient department. The follow-up period for 16 patients was up to 1 year, whereas 9 and 10

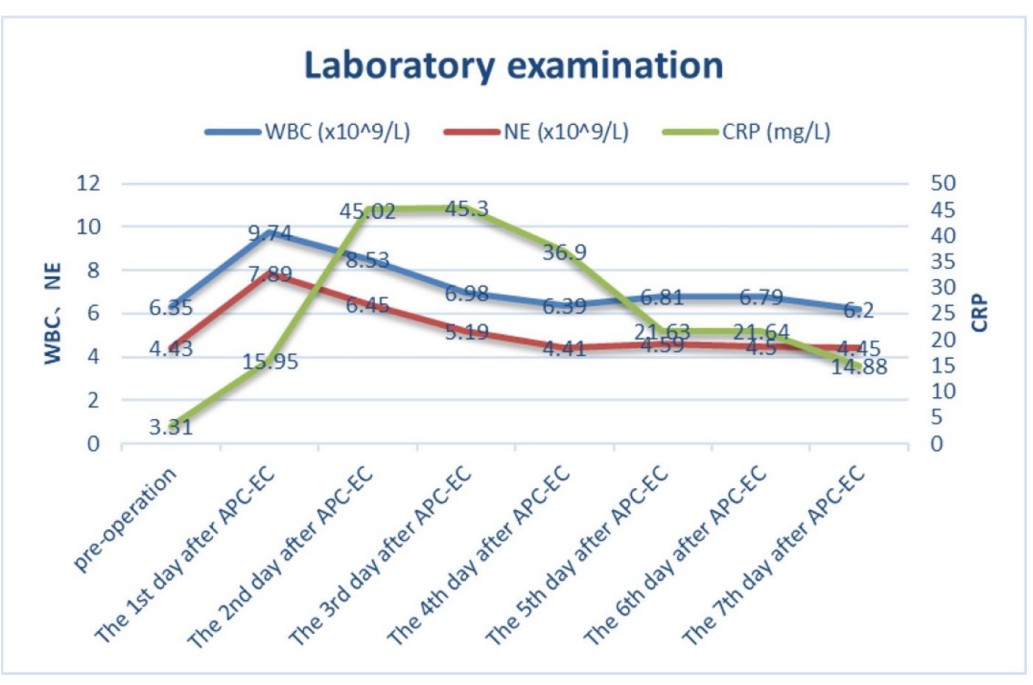

**Figure 2** The trend of white blood cell count, neutrophil count and C-reactive protein before and after endoscopic treatment.

patients were followed-up for 2 or >3 years, respectively. Two patients were found to have late rebleeding, corresponding to the late rebleeding rate of 5.71%. Late rebleeding occurred only in patients diagnosed as having presumptive diverticular bleeding, the cause of which was bleeding from other diverticula. Two patients experienced rebleeding at 2 and 8 months after the initial APC-EC treatment, respectively, but rebleeding was not observed at 3 or 4 years of follow-up after the second APC-EC. The late rebleeding rate at the follow-up periods of 2 and 3 years was 0%. All patients could achieve successful endoscopic hemostasis, bypassing the need for surgery or interventional therapy.

Regarding the clinical manifestations after APC-EC treatment, six (17.14%) patients experienced abdominal pain and distension, and three (8.57%) patients had fever, with the highest body temperature of 38.1 °C. After fasting and treatment with antibiotics, the patients' symptoms alleviated before their discharge, with the average days of postoperative hospitalization being $3.43 \pm 1.51$ days. During the APC treatment, two patients experienced intraoperative perforation. Due to $CO_2$ insufflation and prompt application of endoclips for closure, the patients were discharged on days 3 and 5 after they performed fasting and received antibiotic treatment, respectively. None of the patients required surgery and or experienced delayed perforation.

## DISCUSSION

This study examined the efficacy and safety of APC-EC in patients with CDB. The early rebleeding rate of APC-EC was 8.57%. The incidence of late rebleeding at 1-year follow-up

was 5.71%, whereas it was 0% at the 2- and 3-year follow-up periods. The technical success rate of the new treatment method for CDB was 100%. The rebleeding rate in this study is lower than that reported with conventional endoscopic treatments (*Tsuruoka et al., 2020*). The results confirm that APC-EC is an effective method and associated with mild and manageable adverse events.

CDB usually occurs at the dome or neck of the diverticulum, and because of the damage to the vasa recta in CDB (*Sebastian et al., 2023*), colonoscopy aids in both its diagnosis and treatment. Endoscopic hemostasis can be achieved through epinephrine injection, contact thermal therapy, endoscopic clipping (EC), and endoscopic ligation, which includes endoscopic band ligation (EBL) and endoscopic detachable snare ligation (EDSL) (*Yamauchi et al., 2023*; *Ishii & Imamura, 2024*). Additionally, a study reported that the over-the-scope clip system can also help achieve endoscopic hemostasis (*Wedi et al., 2016*).

Injection therapy is a more suitable treatment option for active bleeding, and it can slow or stop bleeding from the diverticulum (*Kee Song & Baron, 2008*); however, this method has limited efficacy and is associated with a relatively high early rebleeding rate (*Bloomfeld, Rockey & Shetzline, 2001*; *Jensen et al., 2000*). Therefore, injection therapy is often used in conjunction with other endoscopic treatments, such as contact thermal therapy and EC. The early rebleeding rate with injection therapy alone has been reported to be 38% (*Bloomfeld, Rockey & Shetzline, 2001*; *Tanaka et al., 2012*). Thermal contact is a simple and effective method for achieving initial hemostasis in patients experiencing diverticular bleeding (*Jensen et al., 2000*; *Prakash, Walden & Aliperti, 1999*). However, its application is limited by a high rebleeding rate, with the rates of both early and late rebleeding after bipolar coagulation being 40% (*Bloomfeld, Rockey & Shetzline, 2001*). Another retrospective case series reported the early and late rebleeding rates associated with thermal contact therapy as 17%–24% and 0%–40%, respectively (*Strate & Naumann, 2010*). In addition, monopole coagulation therapy in the colonic diverticulum poses the risk of perforation due to the absence of the muscularis propria (*Nagata et al., 2019*).

Currently, EBL and EC are mainly used for managing diverticular bleeding. Most systematic reviews and meta-analyses (*Ikeya et al., 2015*; *Nakano et al., 2015*; *Nagata et al., 2018a*; *Okamoto et al., 2019*) have indicated that EBL is associated with significantly lower rates of early and late rebleeding and long-term recurrence than EC. EC can be applied to the stigmata or feeding vessels to achieve hemostasis through mechanical compression of the vessels (*Kaltenbach et al., 2012*; *Ishii et al., 2012a*). This method causes minimal tissue damage and can be performed using direct and indirect modes. In case the diverticulum is deep or bleeding is heavy, direct EC is not feasible, and indirect EC is used to clamp the bleeding site. Indirect EC involves the use of multiple clips in the form of zipper to clamp the opening of bleeding diverticulum (*Kaltenbach et al., 2012*). The early rebleeding rate of patients with EC had been reported to be in the range 0%–50% (*Jensen et al., 2000*; *Tanaka et al., 2012*; *Kaltenbach et al., 2012*; *Ishii et al., 2012b*; *Strate & Syngal, 2005*). These differences may be related to the mode of endoclip placement onto the vessels. A study reported the early rebleeding rates of 5.9% and 35.7% in the direct EC and indirect EC groups, respectively (*Kishino et al., 2020*). Although direct EC is effective, its clinical

application remains challenging (*Ishii et al., 2012a*; *Ishii et al., 2012b*; *Ishii et al., 2018*), which accounts for its low utilization rate of 24.1% in clinical settings (*Kishino et al., 2020*). The low clinical application of the direct method is attributed to its stringent requirements for achieving stability and expanding the visual field to capture the bleeding vessels. Moreover, the application of the indirect method is associated with a high rebleeding rate. Ishi et al. reported the overall early rebleeding rate of 34% in a cohort of patients treated with EC, among which 85% were treated with indirect EC. Indirect EC often results in incomplete hemostasis because the arcades of the arteries connect at the bottom of the diverticulum (*Ishii et al., 2012b*; *Setoyama, Ishii & Fujita, 2011*; *Shimamura et al., 2016*).

Several studies have documented the efficacy of EBL in achieving mechanical hemostasis of diverticular bleeding (*Nakano et al., 2015*; *Setoyama, Ishii & Fujita, 2011*; *Shibata et al., 2014*). Increasing evidence supporting the effectiveness of EBL in the treatment of CDB has been emerging, with the early rebleeding rate reported to be relatively low (5.6%–15%) (*Ikeya et al., 2015*; *Nakano et al., 2015*; *Ishii et al., 2012b*). EBL has been associated with lower early rebleeding rates than EC. *Nakano et al. (2015)* reported that the cumulative incidence of rebleeding at 1, 12, 24, and 36 months after EBL was 14%, 23%, 26%, and 41%, respectively, whereas the rate of rebleeding after EC treatment was 38%, 49%, 59%, and 68%, respectively. In our study, the cumulative incidence of rebleeding at 1, 12, 24, and 36 months after APC-EC was 8.57%, 5.71%, 0%, and 0%, respectively, which is significantly lower than the corresponding rates reported for EC and EBL. Reduced diverticular rebleeding may be attributed mainly to the treatment of SRH-negative colonic diverticula in APC-EC.

The outcome of CDB was significantly better in the EBL group than in the EC group, possibly because the EBL group experienced a significantly lower rebleeding rate (6%) in the same diverticulum after initial hemostasis treatment than the EC group (22%) (*Okamoto et al., 2019*). Early rebleeding is believed to occur from the same diverticulum; thus, early rebleeding is mainly secondary to inadequate hemostasis (*Nagata et al., 2018a*; *Okamoto et al., 2019*). Conversely, late rebleeding generally occurs from a different diverticulum. Scar formation occurs in 40%–46% of cases involving diverticulum with SRH following EBL (*Nagata et al., 2018b*), which may account for the reduced risk of rebleeding. Although the hemostatic effect of EBL has been recognized, the need for secondary insertion of a colonoscope remains a drawback of this method. In addition, EBL may not be effective in case of firm and large diverticulum due to insufficient suction (*Tsuruoka et al., 2020*).

Unlike EC and EBL, APC-EC is easy to perform and not limited by the shape of diverticulum. Moreover, our results indicate a high technical success rate associated with this method. Studies have highlighted the safety and efficacy of APC in the treatment of various pathologies in the gastrointestinal tract (*Wahab et al., 1997*; *Dumot & Greenwald, 2008*). The core mechanism of APC-EC in the treatment of CDB involves elimination of the diverticular structure. First, APC ablation causes coagulative necrosis of the mucosal layer, resulting in the formation of iatrogenic ulcers (*Nadi et al., 2022*). Subsequently, the ablated mucosal wound is closed using metal clips. This 2-step procedure not only accelerates ulcer healing but also reduces the risk of delayed perforation. The surrounding area of the closed wound is characterized by granulation tissue hyperplasia and scar repair, which eventually

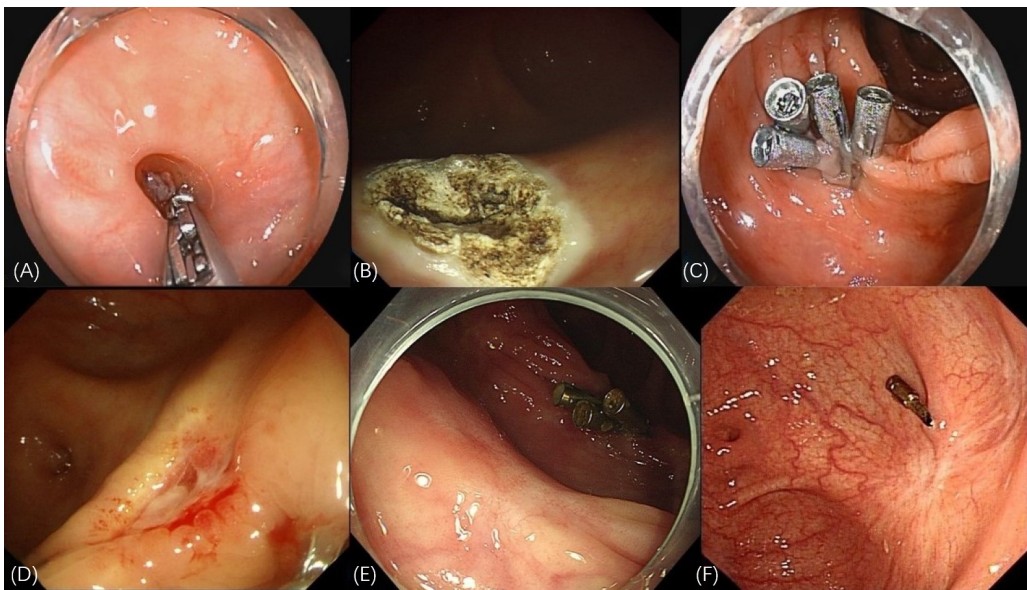

**Figure 3** **Endoscopic manifestations of APC-EC treatment.** (A) The red thrombus at the base of the diverticulum. (B) Thorough ablation of the diverticular mucosa achieved through APC. (C) Multiple endoclips used to seal the wound created through APC cauterization. (D) Premature detachment of the endoclips indicated the formation of an ulcer, accompanied by slight bleeding in the surrounding region. (E) Elimination of the diverticulum, leading to the generation of smooth mucosa, following APC-EC treatment. (F) Formation of white scar tissue, replacing the diverticulum.

form a flat or shrunken mucosal surface. By closing the mucosal layer, this method helps eliminate the diverticular pouch structure, which anatomically interrupts diverticular bleeding. As a new method for the treatment of divercular bleeding, the key points of APC-EC in practical applications are as follows: (1) The dome and neck of the diverticulum should be completely flushed to avoid infection. (2) To ensure the therapeutic effect, the diverticular mucosa should be completely cauterized using APC. (3) The wounded area should be firmly sealed to avoid early rebleeding and postoperative perforation. In this study, two patients experienced intraoperative perforation in the early stage of APC-EC. This perforation was caused mainly by the APC tube that was pressed against the dome of the diverticulum during cauterization, being too close to the mucosa. Hence, while managing CDB, particularly the mucosal layer at the dome, the APC tube must be kept in a "floating" position above the mucosa. In case of intraoperative perforation, timely closure with endoclips and antibiotic administration can reduce the occurrence of peritonitis. Researches have demonstrated that eliminating the target diverticulum with SRH could reduce the late rebleeding rate (*Nakano et al., 2015*; *Nagata et al., 2018b*). In this study, we observed the formation of either scar tissue or newly generated smooth mucosa on the diverticulum surfaces through follow-up colonoscopy (Fig. 3). This outcome prevented the risk of rebleeding through the elimination of diverticula, thereby aiding in attainment of the therapeutic goal.
In the initial stage of APC-EC, the patients experienced rebleeding, which is primarily caused by the early detachment of endoclips. This indicates that the initial treatment is followed by the formation of local ulcers. Therefore, endoclip placement and fully clipping the muscularis mucosa are essential to avoid rebleeding. Another reason for early rebleeding is insufficient mucosal burning of the diverticulum, leading to residual vascular stumps. The patients in this study also experienced late rebleeding, mainly due to diverticulum bleeding at different sites. In such cases, we again applied APC-EC to achieve hemostasis. Although methods for preventing diverticular bleeding have not been standardized yet, we performed prophylactic treatment on diverticula with a high risk of bleeding. None of the patients were found to have rebleeding during the long-term follow-up. However, further studies are needed to ascertain whether APC-EC prophylaxis can reduce diverticular rebleeding in patients with a history of bleeding.

The main clinical manifestations after APC-EC treatment were abdominal pain and fever, which may be attributed to diverticulitis. Based on our experience, we recommend adequately flushing the diverticulum to remove stool as a preventive strategy for diverticulitis. In addition, APC in the colonic diverticulum carries the risk of perforation due to the absence of muscularis propria (*Nagata et al., 2019*). Theoretically, the argon plasma beam follows a path of minimal electrical resistance, which limits its penetration depth (*Farin, 1994*). *In vivo* and *in vitro* experiments indicated that APC created a rather superficial pattern of damage that is suitable for very thin-walled, temperature-sensitive structures, such as cecum and right colon (*Panos & Koumi, 2014*). APC-EC is considered to be safer than the thermal coagulation method. In this method, endoclips are used after APC, thereby preventing postoperative perforation. None of the patients in this study experienced postoperative perforation or required surgical treatment due to rebleeding or perforation.

In Asia, colonic diverticula in the right colon are more common (*Turner et al., 2021*), and many diverticula have a small orifice but a large dome, making it challenging to achieve the desired outcomes using EC and EBL. Unless the diverticulum orifice is large, applying direct EC to the vessels at the base of the diverticulum is unfeasible. At the same time, wide-mouthed diverticula may not be adequately suctioned, leading to the failure of EBL procedures. Compared with EC, APC-EC has minimal requirements on the surgical stability and a higher hemostasis rate. Moreover, APC-EC is superior to EBL in that it avoids re-insertion of the endoscope and therefore suitable in case of multiple or a large diverticulum. It is a simple, easy-to-repeat, and highly maneuverable method, which make its clinical application highly feasible. Single or multiple endoscopic treatments may be effective in eliminating the diverticula and reducing bleeding recurrence.

This study highlights the potential of APC-EC in CDB management; however, acknowledging its limitations is essential. This study used samples from a single center, and the sample size was small. Additionally, the retrospective design of the study makes it challenging to compare the efficacy of APC-EC with those of other methods such as EC and EBL. Although this study provides preliminary data to support the safety, efficacy, and viability of APC-EC in CDB management, large-scale clinical trials are essential to verify its long-term outcomes.

## ACKNOWLEDGEMENTS

I would like to express my sincere gratitude to all the patients who participated in the trials. Their active involvement and cooperation have been of great significance to the development and evaluation of this novel method.

### Funding

This work was funded by the Capital Health Research and Development of Special Fund (Grant No. 2020-4-5123). The funders had no role in study design, data collection and analysis, decision to publish, or preparation of the manuscript.

### Grant Disclosures

The following grant information was disclosed by the authors:
Capital Health Research and Development of Special Fund: 2020-4-5123.

### Competing Interests

The authors declare there are no competing interests.

### Author Contributions

- Zihan Huang conceived and designed the experiments, performed the experiments, analyzed the data, prepared figures and/or tables, authored or reviewed drafts of the article, and approved the final draft.
- Xiaomeng Feng conceived and designed the experiments, analyzed the data, authored or reviewed drafts of the article, and approved the final draft.
- Xin Yin performed the experiments, analyzed the data, authored or reviewed drafts of the article, and approved the final draft.
- Tao Sun performed the experiments, authored or reviewed drafts of the article, and approved the final draft.
- Chongxi Fan analyzed the data, authored or reviewed drafts of the article, and approved the final draft.
- Hongyu Chen performed the experiments, analyzed the data, authored or reviewed drafts of the article, and approved the final draft.
- Bairong Li conceived and designed the experiments, performed the experiments, analyzed the data, prepared figures and/or tables, authored or reviewed drafts of the article, and approved the final draft.
- Shoubin Ning conceived and designed the experiments, performed the experiments, authored or reviewed drafts of the article, and approved the final draft.

### Human Ethics

The following information was supplied relating to ethical approvals (i.e., approving body and any reference numbers):

The Air Force Medical Center granted ethical approval to carry out the study within its facilities (Ethical Application Ref: Air Force Medical Center, PLA (new technique) 2022-17-PJ01).

## Data Availability

Raw data is available in the Supplemental Files.

## Supplemental Information

Supplemental information for this article can be found online at http://dx.doi.org/10.7717/peerj.19910#supplemental-information.

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
