# Peer review of "Innovative utilization of argon plasma coagulation combined with endoclips for managing gastrointestinal bleeding attributed to colonic diverticular bleeding: a retrospective study"

_PeerJ, doi:10.7717/peerj.19910_

## Round 0.1 · original submission · Minor Revisions

Please respond to the comments of the reviewers.

Reviewer 1 ·

Basic reporting

The study was innovative as it solved the problem of natural history of recurent colonic diverticular bleeding.
- However, I am not quite sure that conlcusion of APC and endoclip were safe and effctive is completely correct, as the rate of perforation is almost 6% , eventhough they can be managed with conservative manner).
- The data in the Result was described in a form of very long text, it should be demonstrated in Table.
- As a retrospective design, how come can had the investigator get the inform consents from the patients?

Experimental design

The strength of the study included; most of the patients with presumptive CDB underwent small bowel intervention to enteroscopy and the patients have been followed for an enough time.
I have some comments to improve this manuscript.
- In the presumtive CDB, how do you manage with all diverticula? The rationale of applying APC in each diverticulum, how to select which diverticulum should be sprayed, and the criteria to define adequacy of APC should be described.
- The number of presumtive CDB was two third of the cohort, thus the main focus of this measures were deem to eradicate the diverticula, in which one of them was the culprit of bleeding. The duration and frequency of applying APC in the presumptive bleeding was unclear. The deconstruction of diverticulum is another main outcome of this study. Authors might add more discussion on this rationale of eradication of the diverticulum.
- As the laboratory result of CRP was followed almost everyday, what was the rationale of this monitoring?

Validity of the findings

The conclusion of safety and efficacy of this innovation should be interpret with caution. As the application of APC resulted in perforation in a monority of patients.

Reviewer 2 ·

Basic reporting

1. On page 8, line 72, it might be a clerical error with "CBD", which might be "CDB"
2. It may be better for the latest or newer literature references.

Experimental design

It was well designed, with the Innovative utilization of argon plasma coagulation combined with endocarps. However, it was a retrospective observational study without comparison; it might have been better for a controlled study with a large sample.

Validity of the findings

no comment

Additional comments

no comment

Annotated reviews are not available for download in order to protect the identity of reviewers who chose to remain anonymous.

·

Basic reporting

no comment

Experimental design

no comment

Validity of the findings

no comment

Additional comments

study described a novel treatment modality for colonic diverticular bleeding- endoscopic hemostatsis method using endoscopic argon plama coagulation and endoclips
The methods used are well described and reproducible
The conclusions are representative of objectives of study

---

## Round 0.2 · Minor Revisions

Please correct the manuscript as requested.

**Language Note:** The review process has identified that the English language must be improved. PeerJ can provide language editing services - please contact us at [email protected] for pricing (be sure to provide your manuscript number and title). Alternatively, you should make your own arrangements to improve the language quality and provide details in your response letter. – PeerJ Staff

Reviewer 1 ·

Basic reporting

No further comments

Experimental design

No further comments

Validity of the findings

No further comments

Additional comments

I highly suggest revising the English and grammar. The tense in this manuscript needs to be consistently aligned.

---

## Round 0.3 · accepted · Accept

We are pleased to inform you that your manuscript has been accepted for publication. We look forward to receiving your next manuscript.

With best regards,
Yoshi
Prof. Yoshinori Marunaka, M.D., Ph.D.

Reviewer 2 ·

Basic reporting

-

Experimental design

Looking forward to larger sample-size controlled studies.

Validity of the findings

-